# UNEARTHING LARGE SCALE DOMAIN-SPECIFIC KNOWLEDGE FROM PUBLIC CORPORA

## ABSTRACT

Large language models (LLMs) have demonstrated remarkable potential in various tasks, however, there remains a significant lack of open-source models and data for specific domains. Previous work has primarily focused on manually specifying resources and collecting high-quality data for specific domains, which is extremely time-consuming and labor-intensive. To address this limitation, we introduce large models into the data collection pipeline to guide the generation of domain-specific information and retrieve relevant data from Common Crawl (CC), a large public corpus. We refer to this approach as **Retrieve-from-CC**. It not only collects data related to domain-specific knowledge but also mines the data containing potential reasoning procedures from the public corpus. By applying this method, we have collected a knowledge domain-related dataset named Retrieve-Pile, which covers four main domains, including the sciences, humanities, and other categories. Through the analysis of Retrieve-Pile, Retrieve-from-CC can effectively retrieve relevant data from the covered knowledge domains and significantly improve the performance in tests of mathematical and knowledge-related reasoning abilities.

## 1 INTRODUCTION

Large language models (LLMs) are becoming the new trend not only in natural language processing but also in the entire AI community, pioneered by OpenAI ChatGPT and GPT-4 (OpenAI, 2023). While commercial LLMs are close-sourced, open-source models such as LLaMA (Touvron et al., 2023a) and Mistral (Jiang et al., 2023) are widely studied by the community since they serve as general base models for building LLM applications. Based on these base models, domain-specific models, show great potential in specific domains, such as medicine (Yang et al., 2022; Gao et al., 2023), finance (Wu et al., 2023; Zhang & Yang, 2023), science (Taylor et al., 2022; Wei et al., 2023), and law (Nguyen, 2023; Cui et al., 2023). These domain-specific enhanced models are based on specific human-collected dataset (Azerbayev et al., 2024; Wang et al., 2023c).

However, crafting domain-specific data is very costly. As depicted in Figure 1a, traditional data collection methods involve the selection of relevant resources by domain experts, followed by data collection and processing by engineers. On the one hand, such endeavors are highly labor-intensive, requiring several months of collaboration between multiple domain experts and engineers for corpus collection. On the other hand, some specific domain-related data distribution may be highly scattered, which poses many challenges for large-scale domain-specific data collection. Therefore, in this paper, we introduce an automatic strategy to retrieve data from public corpora for specific domain knowledge, which we call **Retrieve-from-CC**.

In Retrieve-from-CC, we initially collected seed information in some specific domains, such as keywords, frequently asked questions, and textbooks, to serve as inputs for the Query Expanding stage. Leveraging the great generalization capability of LLMs, we can effortlessly expand the initial seed information and extend it to an amount of domain-relevant queries. Inspiration from Wang et al. (2023b) and (Xu et al., 2024), we encompassed two stages of expansion, namely *Question Extension* and *Thought Generation*, which respectively extend the queries in terms of breadth and depth, for retrieving the domain-related data with a broader scope and deeper thought. Subsequently, based on the queries, we retrieved relevant documents from public corpora, and after performing operations such as duplicate data removal and filtering, we formed the final training dataset.

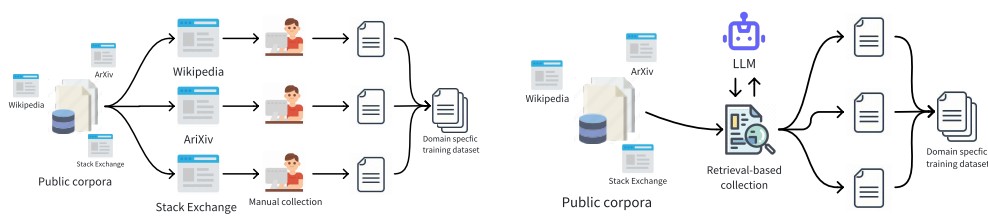

(a) Manual Data Collection      (b) Query and Retrieve Data Collection(our approach)

Figure 1: Comparison of traditional manual data collection methods with our approach.

Leveraging Retrieve-from-CC, we collect a high-quality knowledge dataset Retrieve-Pile, which starts from some seed information of four major domains, including STEM, humanities, social sciences, and medical sciences, as well as general knowledge. Utilizing Retrieve-Pile, we enhance the Llama and Mistral models through further pre-training. Experimental results indicate that through the Retrieve-Pile, Llama and Mistral enhanced models achieved significant performance improvements over baselines in benchmark tests related to mathematics, knowledge assessments, professional examinations, and some complex reasoning tasks.

To sum up our contribution:

- We propose Retrieve-from-CC, a data collection pipeline to retrieve domain-specific knowledge from public corpora, which introduces LLMs to extend query and retrieve domain-related data from public corpora.

- We collect and release a knowledge-related corpora Retrieve-Pile based on Retrieve-from-CC from Common Crawl, a large-scale public corpora, which includes various categories such as STEM, human science, and social science.

- We have analysed the quality and statistical properties of Retrieve-Pile. We examined the distribution of web domain to show the performance of Retrieve-from-CC in collecting scattered information, then, we compared the educational value of Retrieve-Pile with that of other Open-source knowledge-related datasets, to demonstrate the high educational value of our dataset.

- We train several language models on Retrieve-Pile, which demonstrate significant improvements on several professional exams and reasoning datasets.

## 2 RELATED WORK

### 2.1 LARGE LANGUAGE MODEL FOR KNOWLEDGE-BASED REASONING

In recent years, significant progress has been made in the field of Natural Language Processing (NLP), driven by the emergence of large language models (OpenAI, 2023; InternLM-Team, 2023; Bai et al., 2023; Sun et al., 2023). Particularly, in the domain of academic and professional examinations, Some language models such as ChatGPT and GPT-4 (OpenAI, 2023) have demonstrated remarkable success in solving complex tasks, achieving human-like performance through the utilization of the capability of reasoning (Wei et al., 2022; Wang et al., 2023a). However, open-source LLMs lag in performance (Like Llama (Touvron et al., 2023a), Mistral (Jiang et al., 2023) etc.), possibly due to a lack of data.

### 2.2 MANUAL DATA COLLECTION

Extensive efforts are being dedicated to the manually collection of specific training data to enhance the capabilities of Large Language Models (LLM) in knowledge-based reasoning. In the field of mathematics, Lewkowycz et al. (2022) undertook the task of gathering approximately 40 billion tokens of data from arXiv and web math pages. They developed a series of Minerva models based on PaLM (Chowdhery et al., 2023) and observed that augmenting the model with more mathematical data significantly enhanced its proficiency in mathematical reasoning. Similarly, numerous works

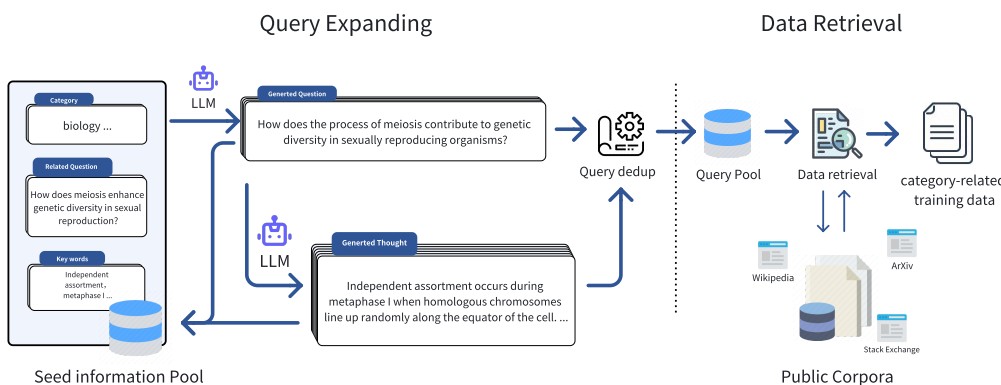

Figure 2: The overview of Retrieve-from-CC's two major components: Query Expanding and Data Retrieval.

(Azerbayev et al., 2024; Wang et al., 2023c; Paster et al., 2024) have undertaken the collection of mathematics-related data, including papers, web pages, and code, at considerable cost.

In the academic and technological domains, Taylor et al. (2022) collected 106 billion tokens of academic and technological data. They asserted that the resulting 120B Galactica model surpasses GPT-3 on various academic benchmarks. These works highlight the effectiveness of manual data collection in enhancing model performance. However, it is crucial to note that these data collection endeavors are labor-intensive, and present scalability challenges, thereby posing some constraints on the overall improvement of model performance.

Overall, manually curated domain-specific datasets typically require substantial human labor for data collection and filtering. For example, the Pile dataset contains 22 distinct web domains, each requiring significant human input for collection, formatting, and initialization. The OpenWebText dataset utilizes multiple filters designed to extract high-quality, domain-specific text. In contrast, the Retrieve-from-CC collects high-quality domain-specific data by simply providing relevant keywords, thereby minimizing the need for manual intervention, reducing human effort required to collect domain-specific data.

### 2.3 RETRIEVAL-BASED DATA COLLECTION

Many works (Li et al., 2023; Li & Qiu, 2023) utilized retrieval methods to enhance their capabilities. The majority of these focus on retrieving documents relevant to the questions to improve the model's prior knowledge, thereby enhancing the performance on knowledge-related tasks and reducing hallucinations. Also, (Yue et al., 2024) retrieved related data for enhancing the instruction synthetic. For data collection, some works (Yao et al., 2022) attempt to use retrieval during the training phase for data collection to improve specified downstream tasks. However, Retrieving specified information for specific downstream tasks relies on the data of those tasks, making it difficult to automate and scale. In contrast, our method introduces LLMs to automatically extend domain-related queries, which enhances the automation and scalability of data collection.

## 3 RETRIEVE-FROM-CC

### 3.1 OVERVIEW

To collect domain-specific data at a lower cost, we propose a retrieval-based method for data collection, which utilizes LLMs to expand keywords into a variety of domain-related queries. These queries are then used to retrieve relevant data from public corpora, the process we refer to as Retrieve-from-CC. An overview of Retrieve-from-CC is illustrated in Figure 2. This framework consists of two main stages: **Query Expanding** and **Data Retrieval**. The input to the entire pipeline consists of seed keywords related to the target domain. These seed keywords are expanded into a broader set of

domain-relevant queries using LLMs during the Query Expanding stage. In the Data Retrieval stage, the generated queries is used as queries to retrieve relevant data from public corpora.

As shown in Figure 2, during the Query Expanding stage, LLMs generate questions and answers centered around the provided keywords. Both the generated questions and the corresponding answers serve as domain-relevant queries that will be used in the Data Retrieval phase. During the Data Retrieval phase, we employ the BM25 algorithm to retrieve documents relevant to the queries, and then we obtain the final data.

## 3.2 Query Expanding

To efficiently and comprehensively retrieve high-quality data relevant to the given seed information, we employed the **Query Expanding**, inspired by Wang et al. (2023b). This phase consists of two steps aimed at broadening the scope of the final query. First, we utilize LLMs to generate questions related to the given keywords or other seed information, we refer it as 'Question Extension.' Subsequently, using these questions, we prompt the LLMs to provide answers and generate reasoning strategies for solving the questions, thereby retrieving thought-related data from public corpora.

**Question Extension**    To expand the scope of our queries, we utilize LLMs to generate questions relevant to the provided seed words. By leveraging the generalization capabilities of LLMs, we can easily generate a series of questions related to the seed information, thereby expanding the conceptual boundaries of the target domain. Figure 2 illustrates an example of 'Question Extension'. Given the seed word 'biology,' LLMs generate related questions, such as those concerning 'genetics' and 'meiosis'. This process evolves limited seed information into a more comprehensive representation that encompasses a range of related concepts. Consequently, this approach significantly enhances the breadth of the query set, ensuring more comprehensive coverage of various aspects within the target domain. All generated questions are saved in the seed information pool for future iterations and also serve as queries for data retrieval.

**Thought Generation**    In addition to expanding the range of queries through Question Extension, we also notice that the answer and the reasoning path are important for ensuring quality. For the generated questions in 'Question Extension', we employ LLMs to generate answers and the reasoning processes required to obtain them. This enables us to acquire detailed and insightful responses. This approach supports a more thorough exploration of concepts related to seed information and facilitates the generation of cognitive processes essential for answering questions. These reasoning processes are also used as queries for data retrieval. While we found that some of these thought data may contain errors or grammatical issues, we still use them as queries to retrieve accurate data from public corpora.

**Post processing**    After generation (both in 'Question Extension' and 'Thought Generation'), the generated data is stored in a seed information pool for the next iteration and also in a query pool for data retrieval. Both processes involve the same post-processing methods: cleaning and deduplication. The cleaning step removes incomplete language data to eliminate the impact of non-natural language. For the deduplication stage, Minhash-LSH (Broder, 1997) was employed to remove duplicate data.

## 3.3 Data Retrieval

Based on the query expanding stage, we get extensive and in-depth queries. During the data retrieval stage, utilizing the enriched queries, we employ the BM25 (Robertson & Walker, 1994) algorithm to retrieve data from general public corpora. BM25 is a widely adopted relevance calculation method commonly used by search engines. It calculates the relevance score between the given query and target documents by weighting and summing the matching degree of keywords in the query with the target documents. Efficiency is the reason why we use BM25 to calculate the relevance. When dealing with billion data, performing relevance calculations for each query against every document becomes exceedingly challenging, while BM25 rapidly retrieves documents relevant to the target query. Compared with Dense Retriever (Karpukhin et al., 2020), it may incur a potential loss in retrieval accuracy, but the latter comes with an unbearable high computational cost. Exploring the potential impact of retriever selection on the quality of collected data might be a valuable direction for future research.

| Datasets | Target domain | automatic | methods | source | Data scale | Open Source |
|---|---|---|---|---|---|---|
| Proof of Pile | Mathematics reasoning | ✗ | human collection | arXiv, Textbooks, Lib., Stack Exchange, ProofWiki, MATH | 8.3B | ✓ |
| OpenWebMath | Mathematics reasoning | ✗ | human collection | Common Crawl | 14.7B | ✓ |
| MATHPILE | Mathematics reasoning | ✗ | human collection | arXiv, Textbooks, Lib., Stack Exchange, ProofWiki, MATH,Web | 9.5B | ✓ |
| AcademicGPT | Academic | ✗ | human collection | arXiv, Unpaywall,Top Universities, Pubmed, Common Crawl, Semantic Scholar, Wiki | 370B | ✗ |
| Galactica | Academic | ✗ | human collection | Papers, Code, Reference Material, Knowledge Bases, Common Crawl, Prompts | 106B | ✗ |
| Retrieve-Pile (ours) | Mathematics reasoning & Knowledge | ✓ | automatic query and retrieve | Public Corpora | 188B | ✓ |

Table 1: Comparation of Retrieve-Pilewith other specific domain knowledge dataset. In this table, most data scales are derived from publicly released research papers, while the data scale for the Retrieve-Pile is obtained through tokenized data analysis using the Llama2 tokenizer.

For each query $q_i$, we conduct the relevance score against every document $d$ in the public corpora $\mathcal{D}$. Followed by sorting the documents based on relevance, we retrieve top-k document set $\mathcal{S}_i = \{\widetilde{d}_1, ..., \widetilde{d}_k\}$ with the highest relevance with the query $q_i$. In our experiments, the typical choice for $k$ is 1000. The retrieved data which related all the query $q_i \in \mathcal{Q}$ is consolidated into training dataset $\mathcal{S} = \cup_i \mathcal{S}_i$.

## 4 Retrieve-Pile

Leveraging Retrieve-from-CC, Based on queries of some knowledge categories, we retrieved several knowledge-related data from processed public corpora. We call the collected datasets as Retrieve-Pile. In this section, we will introduce the analysis of queries (Section 4.1) and the analysis of Retrieve-Pile (Section 4.2). Also, we train several language models to show the improvement of Retrieve-Pile in some knowledge-related reasoning benchmarks (Section 4.3). Otherwise, we discuss about the different when improving different language models using Retrieve-Pile (Section 4.3). See more implementation details in the Appendix.

### 4.1 QUERY ANALYSIS

The progress of query expanding initiates from some categories. Inspiration of the classification of Hendrycks et al. (2021a), we select multiple categories for our initial seed information in the STEM (Science, technology, engineering, and mathematics), Humanities sciences, Social Sciences, and miscellaneous. The key keywords for each category are as follows:

**STEM**: mathematics, physics, chemistry, biology, computer science, engine;

**Humanities**: logical, history, law, philosophy, religions;

**Social science**: econometrics, politics, psychology, sexuality, public relations, psychology, sociology;

**Misc**: medicine, virology, commonsense knowledge.

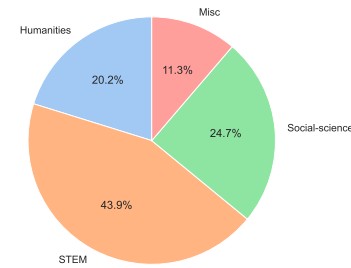

Figure 3: The category distribution of the query for Retrieve-Pile.

After multiple rounds of iterative augmentation and deduplication, we obtain a total of 340,000 queries. The distribution of queries across different domains is depicted in Figure 3. In our query pool, STEM-related queries constitute the majority, while the proportion of queries similar to miscellaneous is relatively small.

### 4.2 DATA ANALYSIS

**Overview**   Based on Retrieve-from-CC, we have formed a high-quality knowledge dataset Retrieve-Pile, which maintains about 735GB disk and 188B tokens (using Llama2 tokenizer). As shown

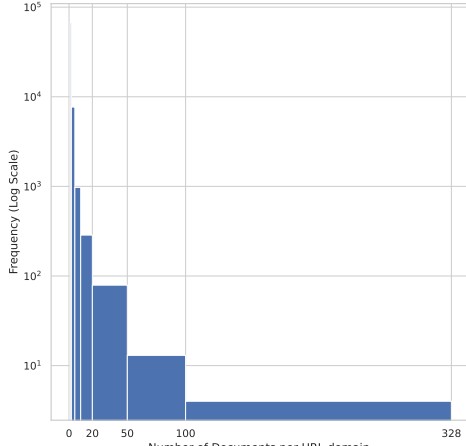
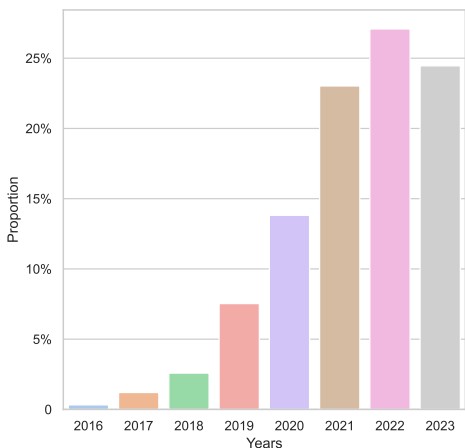

Figure 4: Left: The frequency distribution of the documents number across URL domains, with most domains having few documents, while a small number have many. The y-axis uses a logarithmic scale to highlight this imbalance. this means Retrieve-from-CC not only retrieve the data from high knowledge density websites like Wikipedia but collect data from scatted websites. Right: The timestamp statistics of Retrieve-Pile, most data of Retrieve-Pile come from recent years (different colors represent different years).

in Figure 1, comparing with other datasets in academic and mathematical reasoning domains, we have acquired a large-scale, knowledge-related dataset at a lower cost, without the need for manual intervention. Through automated query expanding, we efficiently capture the information about the seed query. Retrieve-Pile not only covers mathematical reasoning data but also encompasses rich knowledge-oriented corpora spanning various fields such as biology, physics, etc., enhancing its comprehensive research and application potential.

**Web Domain composition of** Retrieve-Pile    Table 2 presents the top 10 web domains with the highest proportion in Retrieve-Pile, which cover a wide range of academic institutions, high-value forums, and authoritative websites in specific knowledge fields. These resources are closely related to the knowledge domains we aim to collect, such as *en.wikipedia.org* and *www.semanticscholar.org*. Many previous research (Touvron et al., 2023a; Gao et al., 2021; Taylor et al., 2022) have specifically collected data from these domains to enrich the knowledge of the training dataset. To gain an insight into the data distribution of Retrieve-Pile, we randomly selected 100,000 examples and conducted the statistical analysis of their domain frequency. Figure 4 left shows the frequency distribution of the document number across the URL domain. We observed that in the Retrieve-Pile, the vast majority of web domains are recorded only once, and these domains also contain rich knowledge content. However, traditional manual data collection methods have limitations in systematically collecting these scattered data, and Retrieve-from-CC has shown its excellent data collection capabilities in this regard.

| Web Domain | Count |
|---|---|
| en.wikipedia.org | 398833 |
| www.semanticscholar.org | 141268 |
| slideplayer.com | 108177 |
| www.ncbi.nlm.nih.gov | 97009 |
| link.springer.com | 85357 |
| www.ipl.org | 84084 |
| pubmed.ncbi.nlm.nih.gov | 68934 |
| www.reference.com | 61658 |
| www.bartleby.com | 60097 |
| quizlet.com | 56752 |

Table 2: Top 10 most web domain of the data in Retrieve-Pile, most of these are academic institutions, high-value forums, and authoritative website.

Furthermore, Table 4 right statistic the timestamps of data sources in Retrieve-Pile by year. It is evident that most of the data in Retrieve-Pile originates from recent years, and the proportion of earlier timestamps is gradually decreasing. This phenomenon can be attributed to the exponential growth of internet data volume and the inherent timeliness characteristic of the Retrieve-Pile.

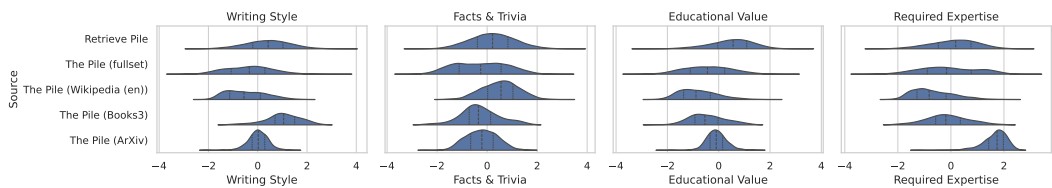

Figure 5: The distribution of QuRating (Wettig et al., 2024) of Retrieve-Pile, The Pile, and selected high-quality subsets of The Pile. QuRating is a robust metric designed to evaluate data quality across four dimensions, with higher scores indicating better quality. Following Wettig et al. (2024), the scores are normalized to have a mean of zero and a standard deviation of one for all displayed data.

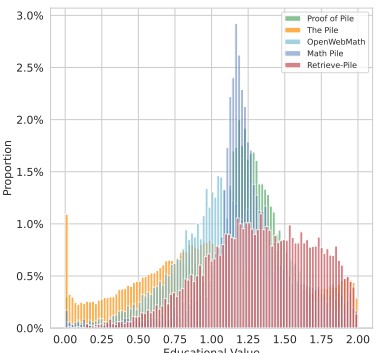

Figure 6: The distribution of educational value of different open source datasets, which shows that the distribution of Retrieve-Pileon the x-axis is significantly right shifted compare others, indicating that it has higher educational value.

| Datasets | Educational Value (↑) |
|---|---|
| MATHPILE (Wang et al., 2023c) | 1.25 |
| Guanaco* (Dettmers, 2023) | 1.115 |
| OpenWebMath (Paster et al., 2024) | 1.089 |
| Proof of Pile Azerbayev et al. (2024) | 1.13 |
| The Pile (Gao et al., 2021) | 1.011 |
| RedPajama* (Computer, 2023) | 0.985 |
| Retrieve-Pile (ours) | **1.291** |

Table 3: The comparison of average educational value scores among different open-source datasets. * denotes the results cited from Tsui (2024), which selected the first 100,000 samples of the dataset, and others randomly selected 100,000 samples.

**Data Quality Analysis**  Data quality is a highly complex concept. Previous studies (Brown et al., 2020; Jiang et al., 2023; InternLM-Team, 2023; Touvron et al., 2023a) labeled certain portions of their datasets as high-quality and employed trained scorers to quantify dataset quality. In our assessment of the data quality in Retrieve-Pile, we leverage utilized (Wettig et al., 2024), which rates the data quality across four dimensions: writing style, required expertise, facts & trivia, and educational value. Also, a 1.3-billion parameter language model was trained using a pair-wise method to assess the quality of the data. Figure 5 illustrates the distribution of QuRating for the Retrieve-Pile and The Pile (Gao et al., 2021), alongside representative subsets of The Pile, including Wikipedia, which scores high on factual, Books3, which exhibits a wide variety of writing styles, and Arxiv, which requires a high level of expertise. As can be seen in the figure, Retrieve-Pile receives high ratings across all dimensions compared to the full Pile dataset. Moreover, in terms of educational value, Retrieve-Pile demonstrates the highest score.

In addition, we employed an open-source data quality classifier[1], to rate the data within Retrieve-Pile. Inspired by Gunasekar et al. (2023), high-quality data should possess characteristics of high educational value, namely: clarity, independence, instruction, and balance. To achieve the assessment of the educational value of the data, Tsui (2024) collected a subset of high-quality raw data and trained a classifier for evaluating the educational value of data based on fasttext [2]. The educational value ranges from 0, indicating low educational value, to 2, indicating high educational value. In Table 3, we present a comparison between Retrieve-Pile and other mainstream knowledge datasets. The results show that Retrieve-Pile has an average score of 1.29, significantly outperforming other open-source knowledge-based real datasets.

Figure 6 further reveals the differences in the distribution of educational value among the Retrieve-Pile, Pile, and OpenWebMath datasets. Through comparison, we can observe a distinct rightward

---

[1] https://huggingface.co/kenhktsui/llm-data-textbook-quality-fasttext-classifier-v2
[2] https://fasttext.cc/

|              |     | MATH  | GSM8K | MMLU  | AGIEval | BIG-Bench Hard |
|--------------|-----|-------|-------|-------|---------|----------------|
| Code-Llama   | 7B  | 3.88  | 14.4  | 40.39 | 21.47   | 42.78          |
| Baichuan2-Base | 7B | 5.72 | 23.81 | 53.95 | 34.68   | 40.32          |
| Minerva      | 8B  | 14.1  | 16.2  | -     | -       | -              |
| Llemma       | 7B  | 14.3  | 35.94 | 47.89 | 24.14   | 48.61          |
| Qwen 2       | 7B  | 10.82 | 51.4  | 57.97 | 40.37   | 22.27          |
| Llama 2      | 7B  | 3.32  | 16.68 | 46.79 | 21.37   | 38.19          |
| Llama 2-QoC  | 7B  | 6.2   | 28.51 | 57.02 | 30.04   | 44.82          |
| Mistral      | 7B  | 11.22 | 47.31 | 64.06 | 32.88   | 56.69          |
| Mistral-QoC  | 7B  | **17.48** | **55.27** | **65.71** | **45.24** | **57.81**  |

Table 4: The performance of our further trained model (Llama 2-QoC and Mistral-QoC) and baselines in some mathematical reasoning tasks and knowledge related reasoning tasks. In this table, all metric is accuracy.

shift in the distribution of Retrieve-Pile, indicating that Retrieve-Pilecontains a greater amount of data with high educational value, while the proportion of low-value data is relatively lower.

## 4.3 THE IMPROVEMENT IN KNOWLEDGE-RELATED REASONING BENCHMARK

For evaluate the improvement of Retrieve-Pile, we conducted two experiments. Firstly, we further trained two models: Llama2-QoC and Mistral-QoC, both of which are based on Llama2 Touvron et al. (2023b) and Mistral Jiang et al. (2023), each with 7 billion parameters. Additionally, we trained a 1.8 billion parameter language model using the Llama architecture from scratch.

**Implementation** In our experiments, we employed the InternLM[3] (InternLM-Team, 2023) library for training all models on 256 A800 GPUs with bfloat16 mixed precision, and only utilized data parallelism during the training process. To enhance throughput and reduce memory consumption, we introduced the Flash attention 2 (Dao, 2024) module. More training details will be described in the Appendix.

During the evaluation, we utilized the open-source library OpenCompass [4], which serves as a platform for evaluating LLMs. Leveraging Opencompass, we compare the performance with some open source pre-trained models: Llama2 (Touvron et al., 2023b), Code-Llama (Rozière et al., 2023), Baichuan 2-Base (Yang et al., 2023), Mistral (Jiang et al., 2023), Qwen 2 (Bai et al., 2023) and some language models for mathematical reasoning: Llemma (Azerbayev et al., 2024) and Minerva (Lewkowycz et al., 2022). For the selection of evaluation datasets, we opted for three distinct capabilities to assess both Llama2-QoC and Mistral-QoC. These encompassed mathematical reasoning datasets such as Math (Hendrycks et al., 2021b), GSM8K (Cobbe et al., 2021), knowledge-oriented language understanding datasets including MMLU (Hendrycks et al., 2021a), AGIEval (Zhong et al., 2024), and challenging reasoning tasks BIG-Bench hard (Suzgun et al., 2023). All metrics of these benchmark reported in this paper is accuracy. More details for evaluation will be described in the Appendix.

**The improvement in further training** The results of our two models trained on Retrieve-Pile (Llama2-QoC and Mistral-QoC) and the baseline are compared in Table 4 across several general benchmarks. Overall, both models exhibit significantly improved performance, particularly Mistral-QoC. In the complex mathematical reasoning benchmark MATH dataset, Mistral-QoC demonstrates a notable enhancement after QoC training, rising from 11.22 to 17.48, which surpasses professional models such as LLEMMA and Minerva by 3 points (14.3 vs 17.48). Furthermore, Mistral-QoC achieves even higher performance on the mathematical application problem GSM8K (47.31 vs 55.27). Turning to various knowledge-based reasoning tasks, Mistral-QoC displays outstanding capabilities in both MMLU and AGIEval. On the challenging BIG-Bench Hard evaluation set, the model also exhibits a noteworthy improvement in handling complex reasoning tasks.

In comparison to the backbone model, LLAMA-QoC and Mistral-QoC show substantial improvements in mathematical and knowledge-based reasoning tests. For instance, Mistral achieves a 6-point improvement in the MATH dataset and a 5-point improvement in GSM8K. In knowledge-based

---

[3]https://github.com/InternLM/InternLM
[4]https://github.com/open-compass/opencompass

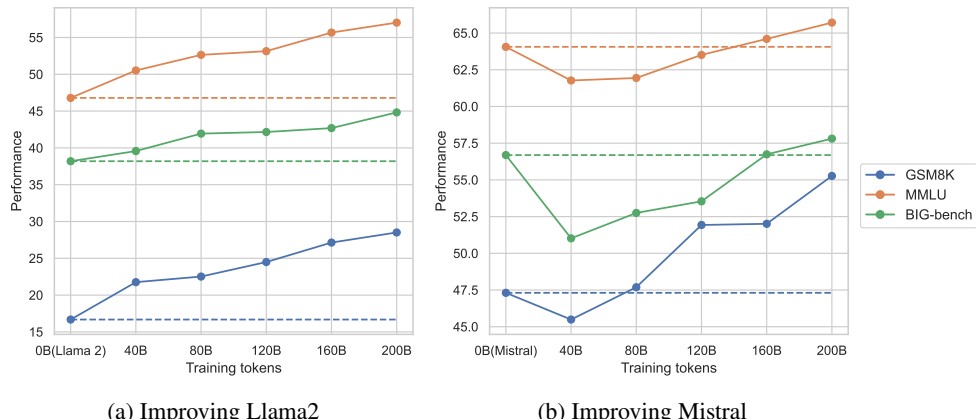

| | (a) Improving Llama2 | | (b) Improving Mistral |

Figure 7: The performance curves of Llama2-QoC and Mistral-QoC, varying with the increase in the number of training tokens.

| | MMLU |
|---|---|
| C4 | 28.52 |
| Retrieve-Pile | 33.13 |

Table 5: The comparison of performace which pre-training in Retrieve-Pile and C4.

| | Overlap Ratio |
|---|---|
| MMLU | 1.5% |
| GSM8k | 0.0% |
| MATH | 0.013% |
| BBH | 0.0% |

Table 6: The data contamination between Retrieve-Pile and downstream task.

tests, MMLU shows a relatively modest improvement, within a 1.7 point. However, in AGIEval, Mistral-QoC outperforms Mistral by an impressive 13 points.

Otherwise, an interesting observation is that when the baseline model performance is lower (e.g., LLAMA2), the enrichment of the dataset leads to higher improvements. Conversely, for baseline models with better performance, achieving significant improvement becomes relatively challenging. This difference may be attributed to variations in the model's ability to fit the data.

**Difference between Improve Llama and Mistral**   As shown in Table 4, it is evident that Llama2 exhibits inferior performance relative to Mistral. However, this also highlights a greater potential for performance improvement when training in Retrieve-Pile. Also, we observe a significant behavioral difference during the improvement process, which is shown in Figure 7. We found that in certain tasks, Mistral's performance undergoes a certain degree of decline during the improvement process, followed by a subsequent ascent after some time, eventually surpassing the previous performance levels. This phenomenon may come from a conflict between the potential distribution of the model and the distribution of high-quality datasets. Mistral's potential distribution is superior but more unstable, whereas Llama's performance is relatively poorer but exhibits greater plasticity. On the other hand, the improvement achieved on more powerful models (Mistral 7B) also demonstrates the high-quality of Retrieve-Pile.

**Comparation with other pre-training data**   Similarly, we evaluated the impact of Retrieve-Pile on model performance during the pre-training phase. Table 5 compares the performance of a 1.8B-parameter language model pre-trained on 200B tokens from Retrieve-Pile with that of C4 (Raffel et al., 2020) (an open-source pre-training dataset curated and filtered from Common Crawl) on MMLU. The 200B tokens for C4 were randomly sampled. The model trained on C4 achieved only 28.52% accuracy on MMLU, whereas the model trained on Retrieve-Pile showed a 5% point improvement (33.13%) in MMLU compared to C4, which highlights the advantage of Retrieve-Pileover randomly sampled data.

**Data contamination**   Data contamination analysis is equally important in the study of pre-training corpora. Many research (Chowdhery et al., 2023; Touvron et al., 2023b; Jiang et al., 2024) have

defined data contamination as the n-gram overlap between the pre-training corpus and the test set. For instance, PaLM Chowdhery et al. (2023) defines contamination as a 70% 8-gram overlap, while Llama2 Touvron et al. (2023b) considers it contamination if more than 10 tokens overlap between the training and test sets. Inspired by previous works, we utilized the Overlapy codebase to calculate the 8-gram overlap between Retrieve-Pile and the test set. Table 6 shows the proportion of documents with 8-gram overlaps between Retrieve-Pile and downstream tasks. It is important to note that this metric calculates direct overlaps, potentially leading to false positives, as overlaps may occur between semantically different contents. As shown in table 6, the overlap across most downstream tasks is relatively minimal, generally below 0.1%. For example, gsm8k and BBH exhibit an overlap rate of 0%, while MMLU, containing a significant amount of conceptual content, shows only about a 1.5% overlap. One potential explanation is that MMLU incorporates substantial knowledge-related questions, particularly general knowledge, which may be reflected by similar descriptions in the pre-training corpus. Additionally, domain-specific knowledge typically consists of specialized terms and standardized expressions, which are prone to repetition. Overall, the low n-gram overlap of Retrieve-Pile across downstream tasks suggests that contamination in Retrieve-Pile may not be significant. This also highlights the dataset's broad adaptability in handling diverse tasks.

## 5 LIMITATION AND HALLUCINATION ANALYSIS

### 5.1 LIMITATION

In this paper, we propose a collection method for domain-specific data, Retrieve-from-CC, which extends domain-relevant queries through LLMs generated for data retrieval. Additionally, we demonstrate through empirical evaluation on further training of LLAMA and Mistral that data collected using this method significantly improves the ability of LLM in some specific domains. However, this method still has the following potential limitations:

**Data Quality** The quality of data collected by Retrieve-from-CC depends largely on the quality of data in public corpora. In this work, we utilized the Common Crawl corpus as our public corpus, extracting and processing the CC dump up to April 2023. Despite leveraging some methods (Wenzek et al., 2020) for processing high-quality web data, we found that the corpus still contains an amount of low-quality and erroneously extracted content. Thus, improving the data quality of public corpora is also a direction for future research.

### 5.2 HALLUCINATION ANALYSIS

The output of LLMs generally exhibits significant hallucination issues. Previous work has shown that LLMs are prone to hallucinations, and using their output directly in training without filtering can exacerbate the model's hallucinations for certain issues. However, in the data collection pipeline of Retrieve-from-CC, the model is only used to generate queries and is not directly applied in training. Therefore, even if the synthesized queries from the LLMs contain incorrect information, the information retrieved from the corpus based on these incorrect queries is correct. Hallucinatory queries do not lead to the retrieval of incorrect information.

## 6 CONCLUSION

In this study, we propose an efficient method Retrieve-from-CC, for the automated collection of specialized domain data. Leveraging seed data from some specific domains, we employ a language model for query expanding. By optimizing the breadth and depth of queries, we expand the query to retrieve data relevant to the specified domain. Ultimately, we collected and released an open dataset comprising approximately 735GB of Retrieve-Pile, equivalent to approximately 188 billion tokens, in the fields of mathematics and knowledge. Experimental results demonstrate that the adoption of Retrieve-Pile significantly enhances the model's performance in some reasoning tasks, such as math word problems and professional examinations. Our objective is not only to establish a research foundation for community studies in mathematical and knowledge-related reasoning but also to provide an efficient and cost-effective method for collecting high-quality data, thereby facilitating the accumulation of more high-quality data.

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

## A  MODEL TRAINING DETAILS

In this study, we train all models with bf16 mixed precision and only data parallel. We employed the AMSP (Chen et al., 2023), shard optimizer state across 8 cards to reduce communication overhead. Simultaneously, data parallelism was employed for training. To enhance throughput and reduce memory consumption, we introduced the Flash attention 2 (Dao, 2024) module. All models underwent training for 50,000 steps, with a global batch size of 4 million tokens per step, totaling 200 billion training tokens.

During the initial 2,000 steps of training, the learning rate was warmed up to the maximum, and then, at the end of training, it was decayed according to cosine decay to the specified minimum learning rate. Specifically, for Llama2-QoC, the maximum learning rate during training was set to 2e-5, and the minimum learning rate was set to 2e-6. For Mistral-QoC, the maximum learning rate during training was 5e-6, descending to 2e-7 at the end of training. In our experiment, Llama2-QoC and Mistral-QoC achieved a training throughput of approximately 4000 tokens per GPU per second(TGS). Despite the relatively higher training efficiency of Llama2-QoC, it still utilized 14,000 GPU hours.

## B  EVALUATION

During the evaluation, we utilized the open-source library OpenCompass [5], which serves as a platform for evaluating large models. OpenCompass offers various evaluation datasets and supports efficient task partitioning to maximize the utilization of computational resources. For the selection of evaluation datasets, we opted for three distinct capabilities to assess both Llama2-QoC and Mistral-QoC. These encompassed mathematical reasoning datasets such as Math, GSM8K, knowledge-oriented language understanding datasets including MMLU, AGIEval, and challenging reasoning tasks BIG-Bench hard. The details of evaluation datasets are as follows:

**Math (Hendrycks et al., 2021b)**   Math datasets comprising 12,500 competitive mathematical problems spanning challenging areas such as algebra and number theory. During evaluation, we selected four questions as examples, and each illustrating complete steps of problem-solving approaches. We assessed the model-generated answers for equivalence with these golden answers.

**GSM8k (Cobbe et al., 2021)**   GSM8k datasets contain 8.5k high-quality grade school math word problems, the task requires the large language model answered to combine world knowledge and mathematical reasoning. In this task, following xxx, we provide four questions and the solution with more detail as examples and also evaluate the equivalence of generated answer with the golden answer.

**MMLU (Hendrycks et al., 2021a)**   MMLU is a vast multi-task dataset encompassing questions from various disciplines, including humanities, social sciences, STEM, and others. There are 57 sub-tasks in MMLU including elementary mathematics, American history, computer science, law, etc. During the evaluation, we provided five example questions and their answers with relatively complete chains of thought, using this to judge whether the model could correctly select options. This task necessitates a broad range of world knowledge and problem-solving capabilities for large language models.

**AGIEval (Zhong et al., 2024)**   AGIEval is also a benchmark that has various categories, and is designed for accessing the performance of large language models in context with human-centric standardized exams. Compared to MMLU, this dataset has more comprehensive data sources and provides a better evaluation of cross-linguistic knowledge performance.

---

[5]`https://github.com/open-compass/opencompass`

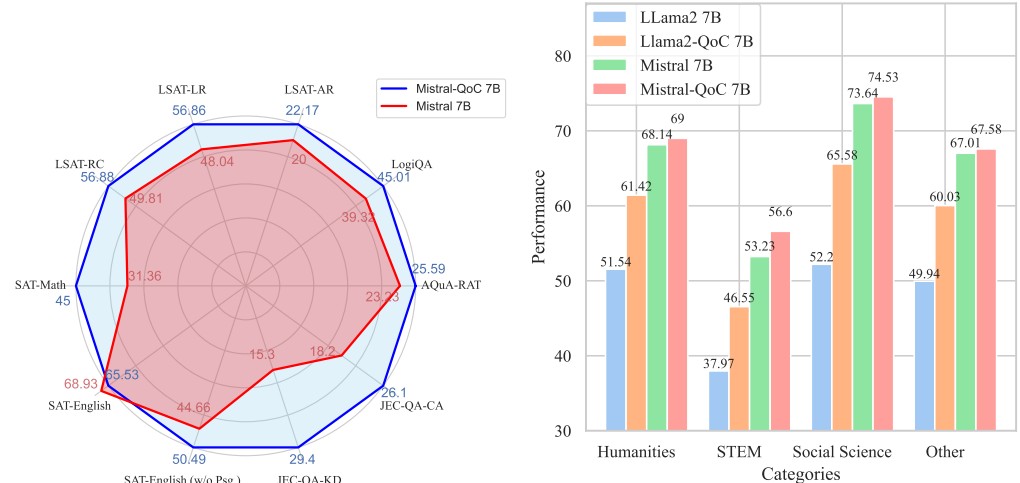

Figure 8: Left: the comparison of performance in AGIEval Benchmark between Mistral and Mistral-QoC. Right: The performance comparison of performance in MMLU.

**BIG-Bench hard (Suzgun et al., 2023)**   BIG-Bench hard is a challenging subset of BIG-Bench, which is designed to evaluate the reasoning ability of large language models. This dataset comprises 23 BIG-Bench tasks covering different programming languages and domains. During the evaluation, we provide three examples of large language models and assess whether the model could accurately answer the presented problems.

## C   MORE EVALUATION OF LLAMA-QOC AND MISTRAL-QOC

In order to compare the effects of Retrieve-Pile on knowledge-related reasoning abilities, Figure 8 contrasts the improvement based on Llama2 and Mistral on MMLU, with a specific focus on performance across different subjects.

From the perspective of Llama-QoC and Llama, following training with Retrieve-Pile, the average performance on MMLU for all categories increased by at least 10 points. This notable performance enhancement can be attributed in part to the initially lower performance of Llama, while the Retrieve-Pile exhibits relatively rich knowledge content, playing a significant role in driving the improvement of Llama2's performance.

In comparison to Llama, Mistral's performance improvement is relatively moderate. The results indicate that in the STEM field, Mistral-QoC demonstrates a higher performance improvement compared to other fields, rising from 53.23 to 56.6. In contrast, performance improvements in other categories hover around one point. Two possible reasons account for this phenomenon. Firstly, Mistral exhibits strong comprehension abilities in disciplines such as humanities and social sciences, consistently scoring above 65, leaving relatively limited room for improvement. In contrast, the model's score in the STEM field is only 53.23, suggesting a greater potential for performance enhancement. Secondly, Retrieve-Pile has the largest and most extensive dataset in the STEM field, contributing to the more pronounced performance improvement of Mistral-QoC on MMLU-STEM.

Figure 8 compares the Performance of professional and academic exams covered in AGIEval. Even in the case of professional academic exams, Retrieve-Pile similarly leads to substantial performance improvements. As observed in the figure, apart from a slight decline in the SAT-English expression test (from Mistral 68.93 to 65.53), other tasks exhibit noticeable improvements after continued training in Retrieve-Pile. Notably, SAT-Math rises from 31.36 to 45, which is a significant improvement for Mistral. Legal exams such as JEC-QA-CA and JEC-QA-KD also show significant enhancements, with the performance increasing from 18.2 and 15.3 to 26.1 and 15.3. For other law-related exam, LSAT is divided into three aspects: Law-Analytics (LSAT-AR), Law-Logic(LSAT-LR), and Law-Reading(LSAT-RC) improvements of 2.17, 8.82, and 7.07 point, respectively. For the LogiQA,

Mistral-QoC achieves an accuracy of approximately 45.01%, nearly a 20% improvement compared to Mistral (39.32)

## D   COLLECTION IMPLEMENT DETAILS

In the implement section, we primarily discuss three components, the public corpora (D.1), retrieval engine (D.2), and the post-processing settings (D.3).

### D.1   PUBLIC CORPORA

With the development of large language models, public corpora have become increasingly rich, including the Pile, RedPajama, and Common Crawl. Common Crawl is an open-source web crawler project containing all publicly available web pages from 2013 to the present. In theory, it encompasses a significant portion of the information present on the web. Due to hardware cost constraints, we utilized WARC format data from the years 2016 to 2023 to build our retrieval database. We performed extraction, filtering, and cleaning procedures on the data obtained from Common Crawl to ensure the quality of the retrieval dataset. The final retrieval database comprises several billion records, occupying a total of 50TB of disk space.

### D.2   RETRIEVAL ENGINE

Retrieving data from billions of documents is highly challenging, and we need to calculate the relevant score between queries and each document in the retrieved database. To improve storage and retrieval efficiency, we built the retrieval engine based on Elasticsearch (ES). Elasticsearch is an open-source distributed search engine that employs distributed storage and inverted indexes, achieving data retrieval highly efficient. We selected the BM25 algorithmic as the relevance scorer because of its efficiency. For each query, we recall the top 1000 most relevant documents. Leveraging Elasticsearch's efficient storage and indexing algorithms, we can complete a query retrieval within 100ms.

### D.3   POST PROCESSING

To enhance the quality of Retrieve-Pile, inspired by Wenzek et al. (2020), we conducted data quality filtering and deduplication in the post-processing stage.

In the data quality filtering phase, we manually selected some high-quality data as positive examples and randomly selected low-quality data from Common Crawl, such as poorly structured data and advertising data, as negative examples to train our scoring model. High-quality data mainly includes papers, books, and high-quality forum data. We chose BERT-base-uncased as the backbone to train the model and tested it on a subset of data to ensure its high usability. Finally, we scored the retrieved data and filtered out data with quality scores below 0.8.

In the deduplication phase, we employed the Minhash-LSH method. For hyperparameter selection, we computed the similarity scores based on 13 grams and set the similarity threshold to 0.8. Additionally, we set $num\_perm$ to 128 to balance computation efficiency and deduplication performance.

## E   PROMPT OF QUERY EXPANDING

================== PROMPT OF QUESTION EXTENSION ==================

Suppose you are a question creator and your task is to create a new question based on the example question!
Note that the new questions should have the same domain as the example questions, but be less frequent, of
exactly the same length and difficulty as the example questions. You need to use your ingenuity to create a
problem that is completely different from the given problem.
Sample questions will be given after ###Given Question###. You need to write the newly created question
after ###Created Question###.
###Given Question###

[question]

================== PROMPT OF THOUGHT GENERATION ==================

Suppose you are a expert and your task is answer the given problem and tell me how to get the answer!
You need to write the answer after the ###Answer### symbol. Please write the chain of thought after the
###COT### symbol.
###Given Question###

[question]

