# OpenReview forum: "Unearthing Large Scale Domain-Specific Knowledge from Public Corpora"
_ICLR.cc/2025/Conference — Submitted to ICLR 2025_

### Official Review · Reviewer_cx4Y · 2024-11-01

**Soundness:** 2
**Presentation:** 1
**Contribution:** 2
**Rating:** 5
**Confidence:** 4

**Summary:**

This work presents an automated pipeline to collect domain-specific data from public corpora by leveraging large language models (LLMs) for query expansion and BM25 for efficient retrieval. The resulting dataset spans multiple domains, including STEM, humanities, social science, and Misc. While the method emphasises scalability and cost-effectiveness over manual curation, it also faces challenges in ensuring data quality and distinctiveness compared to existing human-curated datasets. Experimental results indicate that models trained on the proposed dataset are improved in several reasoning benchmarks.

**Strengths:**

- This work addresses a well-motivated goal: developing an accurate and scalable approach to extract training data from the evolving web, which is essential for keeping LLMs up-to-date.

- Experimental results show clear improvements in LLM performance on the listed benchmarks when further trained on the proposed dataset.

- The authors provide a detailed analysis of the proposed dataset, including multiple evaluation factors. Notably, the data leakage analysis between the pre-training dataset and the evaluation benchmarks is a valuable addition, helping to ensure the integrity of the results.

**Weaknesses:**

### About Novelty

- The authors have not clearly demonstrated how the proposed dataset differs from or adds unique value compared to existing high-quality, human-curated datasets, including the ones shown in Table 1.

- Additionally, the proposed pipeline lacks distinctive features that would make this automated construction process stand out as an innovative or superior alternative. For example, since this work focuses on domain-specific knowledge, it can be beneficial to leverage knowledge bases  or other kinds of structured data to help improve the relevance and accuracy of the data points.

### About Methodology
- While the authors emphasise scalability, fully relying on LLMs to refine and expand queries is both computationally expensive and prone to errors.

- Although BM25 offers efficiency and scalability in the retrieval phase, it does not guarantee high accuracy in the retrieved `(query, answer)` pairs. Even if standard dense retrieval techniques were employed, achieving consistently high accuracy would remain challenging.

- Consequently, errors introduced during both the query generation and retrieval phases could propagate, potentially compromising the overall quality of the final dataset.

### About Evaluation

- Given that LLM pre-training typically involves a broad set of evaluation benchmarks, this work lacks an analysis of potential "forgetting" on benchmarks (e.g., HELM, GLUE, LLM leaderboard) not included in the listed experiments.

- When comparing with other pre-training data, if you aim to demonstrate that this automatically generated dataset holds value even against human-curated datasets, it would be essential to directly compare their performance. Ideally, this comparison would show that the proposed dataset lags only by a small margin. Alternatively, comparing it with well-established synthetic data generation methods (also discussed in Suggestion 1) would also help substantiate the dataset's value and highlight areas for potential improvement.

**Questions:**

### Suggestions

- Given the fully automated nature of the proposed pipeline, the resulting dataset is more like a synthetic dataset. This raises a different research question that how to generate high-quality synthetic datasets effectively and how this method compares to existing synthetic data generation approaches.

- I recommend that the authors invest effort in refining the paper's writing, as several language issues currently affect the fluency of reading.

- I recommend ensuring consistency in terminology throughout the paper. For instance, terms like 'Retrieve-Pile' and 'Knowledge Pile' appear to refer to the same dataset, which could cause confusion.

---

> ### Author Response · Authors · 2024-11-21
>
> Very thank you  for your helpful comments. Below, we will explain your concerns in detail:
>
> About Novelty
>
> 1. We apologize for not elaborating on the differences between our method and other manually curated data collection approaches. In the revised version, we have provided a detailed comparison:
> The primary distinction between the Retrieve-from-CC and manually curated datasets lies in the reduction of human effort required to collect domain-specific data. Manually curated domain-specific datasets typically demand substantial human labor for data collection and filtering. For instance, the Pile dataset contains 22 different web domains, each requiring considerable human input for collection, formatting, and initialization. The OpenWebText dataset employs multiple filters designed to extract high-quality domain-specific text. In contrast, the Retrieve-from-CC collects high-quality domain-specific data by simply providing relevant keywords, thereby minimizing the need for manual intervention.
>
> 2. I have some confusion regarding the term "distinctive features" that you mentioned. Could you clarify what specific aspects you are referring to? Our method requires only a few keywords to collect domain-relevant data. While collecting better domain-specific datasets as seed data, as in your example, could indeed enhance the query's quality and relevance. Also, it would increase human costs. Moreover, this is not the primary focus of our paper.
>
> About Methodology
>
> 1. As mentioned in our paper, in our approach, LLMs only generate the query and do not directly affect the quality of the final data. Moreover, compared to manual data collection at this scale, model generation is not costly.
>
> 2. In our paper, the retriever's role is solely to collect relevant documents, and there is no necessity to gather extremely precise data. Of course, a better retriever could fetch more relevant data. However, as noted earlier, as similar as data sources , and this is not the primary focus of our paper. Nevertheless, the points you raised are crucial for refining the data, and we will continue to improve the process in future work.
>
> 3. We have discussed in the paper the potential risks of hallucination in LLMs. Since our data is retrieved from publicly available corpora, the hallucination issue does not directly affect the quality of our data.
>
> About Evaluation
>
> 1. We acknowledge the extensive benchmarking involved in the pre-training process. However, since the core focus of this study is to analyze the effectiveness of Retrieve-Pile, the "forgetting effect" was not included in this experiment. We will discuss the benchmarking results related to forgetting in future works.
>
> 2. We understand the importance of comparing auto-generated datasets with manually curated datasets. To this end, we have compared the "educational value" and "QuRating" of our data with those of manually curated datasets in the data quality analysis section. This comparison contrasts the data quailty of Retrieve-Pile and human-curated data. While model-based evaluations are also valuable, we can include a direct comparison with human-curated datasets in future versions and demonstrate the performance differences. Although we acknowledge the merit of comparing with other synthetic data generation methods, we have not included this in the current paper. We consider the data we collect to be real data, not synthetic. Nevertheless, we will add comparisons with related synthetic data work in subsequent revisions to clarify this issue.
>
> For Suggestions
>
> 1. We apologize for the confusion; however, our work is not directly related to synthetic data. The primary motivation of this paper is to leverage the generalization capabilities of large models to generate queries based on domain-specific keywords and retrieve domain-relevant data, which is essentially a data collection task. We will carefully consider this point in future revisions and provide a discussion comparing synthetic data.
>
> 2. We sincerely apologize for any inconvenience caused to your reading. We are currently revising the paper to improve its clarity and readability.
>
> Once again, we appreciate the time and effort you spent reviewing our paper and providing valuable feedback.

---

> ### Comment · Reviewer_cx4Y · 2024-11-21
>
> Thank you for your response. Here are some immediate thoughts after reading it (I will post more if any):
>
> **Regarding your comment: "I have some confusion regarding the term "distinctive features" that you mentioned."**
>
> To clarify my earlier point about "distinctive features," I was referring to the mechanisms or methods you employ to ensure the actual relevance of the datasets. For example, leveraging knowledge bases is just one potential approach for verification, but my suggestion is not limited to this method. The key question is: what features or processes are in place in your framework to ensure the generated datasets are actually relevant to the domains?
>
> **Regarding your comment: "Our work is not directly related to synthetic data."**
>
> Upon closer examination of the paper, it seems that the final data points come from public corpora rather than being generated by LLMs. If this understanding is correct, then the dataset indeed would not fall under the category of synthetic data. I would consider the points regarding synthetic datasets addressed.
>
> **Regarding your comment: "In our approach, LLMs only generate the query and do not directly affect the quality of the final data."**
>
> I believe this statement may not be fair. LLMs are responsible for generating "anchors" used to retrieve or select useful documents. Since this step plays a critical role in shaping the final dataset, any irrelevant or low-quality information generated during this process could potentially impact the overall quality (especially the relevance) of the data.
>
> **Further questions**:
>
> - Could you clarify the origin of the keywords in the seed information? Are they manually curated, automatically generated, or derived from another source?
>
> - Another point of confusion is that this work claims to be domain-specific, yet the collected datasets encompass several topics. Nevertheless, I do appreciate the authors’ response to reviewer 8mWq to include at least an analysis focused on a particular domain.
>
> **A further suggestion**:
>
> Since the ultimate datasets are essentially subsets of existing pre-training datasets, further training on these subsets does not inherently guarantee better quality or performance. To validate this claim, it would be beneficial to include a comparison. For instance, you could evaluate LLMs trained on data points that are excluded from these subsets (not necessarily the entirety of the excluded data due to scale, but a reasonable portion).

---

> > ### Author Response · Authors · 2024-11-22
> >
> > Thank you for your timely response. Below, we will answer your questions in detail:
> >
> > 1. What features or processes ensure that the generated datasets are relevant to the target domains in your framework?
> > This study primarily focuses on generating relevant queries using a limited set of keywords and retrieving high-quality, domain-specific documents from public corpora. While utilizing high-quality knowledge bases could enhance the relevance of retrieved documents, this is not the primary focus of our work. Our approach does not impose strict domain limitations on the generated datasets. However, the results demonstrate that our dataset significantly improves performance in the target domains (as shown in Section 4.1). Incorporating knowledge bases or exploring methods to analyze and constrain the relevance of retrieved data to target domains could further enhance our approach. We plan to address this aspect in future work.
> >
> > 2. Origin of Keywords in the Seed Information:
> > The initial seed information was initialized based on the classifications proposed by Dan Hendrycks et al. Keywords were derived from these categories, supplemented with others manually collected from the internet. The initial number of keywords was limited,  the expansion of the keyword set primarily relied on iterative processes involving question extension and thought generation. The generated texts were added to the seed information database, enabling further iterations to enrich the query.
> >
> > 3. Domain-Specificity and Coverage of Several Topics:
> > We believe that "domain-specific" and "covering several topics" are not mutually exclusive. Our primary focus is on enhancing datasets for the domains highlighted in Hendrycks et al. (2021a), which have lacked open-source datasets specifically constructed for these areas. Our dataset addresses this gap. Additionally, we evaluated our approach using the BIG-Bench Hard dataset, which focuses on complex reasoning tasks. We observed that training on large-scale knowledge-based data significantly improved the model's performance on such tasks.
> >
> > We sincerely appreciate the reviewer’s insightful feedback. Should you have further questions, we will respond promptly.
> >
> >
> > [Hendrycks D et al.] Measuring massive multitask language understanding

---

> > > ### Comment · Reviewer_cx4Y · 2024-11-23
> > >
> > > Thank you for your responses. After reviewing the rebuttal, I would consider this work to be a borderline case.
> > >
> > > On the positive side, the paper adopts a reasonable pipeline for retrieving documents from public corpora, which leads to improved performance on certain benchmarks. However, there are a few aspects that could be strengthened:
> > >
> > > - The writing, as noted in my initial review, has caused several points of confusion. While the revision has addressed some of these issues to a certain extent, further improvements in clarity would benefit the paper. For example, it would be good to see the  examples to illustrate how the proposed approach retrieves reasonable and relevant documents from public corpora.
> > > - The retrieval criteria, while functional, are not entirely convincing. That said, I understand that it may not be feasible for the authors to explore more comprehensive criteria within the constraints of the rebuttal phase.

---

### Official Review · Reviewer_U334 · 2024-11-03

**Soundness:** 3
**Presentation:** 1
**Contribution:** 2
**Rating:** 3
**Confidence:** 3

**Summary:**

The authors propose Retrieve-from-CC, a data collection pipeline consisting of a a) query generation process and b) a document retrieval process based on the generated queries. They argue that their proposed method makes it possible to automate the data collection process for high quality domain-specific data.
The so created dataset is then evaluated with regard to its composition (sources and domains) and its data quality (quantified as QuRating).
Finally, the authors use their newly composed dataset to fine-tune two LLMs (based on Llama2 and Mistral) and to train a Llama model from scratch. These models are then evaluated with regard to their performance on various benchmark datasets on mathematical and knowledge oriented language understanding tasks.
Their experiments indicate that their newly collected dataset of domain-specific high quality data can be used for fine-tuning LLMs and improve their performance on tasks that require specific knowledge. They additionally show that there is little to no dataset contamination when comparing the downstream task data with their own data.

**Strengths:**

+ extensive experiments, showing that their proposed data is helpful for training “smaller” (i.e. 7B Parameter) LLMs on domain specific task
+ automating the process of curating high-quality domain specific data
+ addressing the issue of data contamination

The authors propose a method of automating the process of curating high-quality data in specific domains. Furthermore, they showcase how their data can be used to fine-tune LLMs and improve their performance on benchmark tasks within the respective domain.

**Weaknesses:**

1) writing and language throughout the paper needs improvement. This makes the present work difficult to follow at times:

l. 67 “Otherwise, […]” -> additionally?
l. 70 “[…] continuing learning […]” -> unclear if they talk about fine-tuning
l. 82 “[…] for the quality and statistical of Retrieve-Pile […]” -> statistics?
l. 82 “We statistic […]” -> usage as a verb?

Paragraph starting at l. 179: Is question evolution the same as question extension from the previous paragraph?

l. 237 “Otherwise, we discuss about the different when improving different […]”

Unfortunately, these are only some highlighted examples where the quality of presentation is lacking.

2) exposure of results:

Table 4: It would be beneficial to mention the reported metrics. If not here, then at the description of the benchmark datasets.

This issue is persistent in most of the evaluation section. The authors report an increase of performance in “points” on multiple occasions, which frankly speaking could mean anything.

In the case of data quality, a little more in-depth explanation of the QuRating would be helpful in understanding the results. As this is by now not a wide-spread metric, it would be beneficial to explain how the values are obtained and what exactly they mean.

Overall, the authors could improve the exposition of results by providing a more detailed explanation of the used metrics, as this is an important bit of information for the reader.

3) focus:
Overall, the focus of the paper is not very well defined. First, the authors introduce a dataset collection method and provide a detailed overview of the collected dataset. For the present work to be a resource paper, the proposed Query Bootstrapping methods is not explained in sufficient detail. The paragraphs on “Question Extension”, “Thought Generation” and “Query post processing” are rather vague.

The other side of the spectrum would be a paper on an empirical study. For this to be the case, the evaluation section would require to be more detailed (regarding reported metrics).
In addition to that, following one of the author’s main argument (automating the process of collecting high-quality domain-specific data), it would be nice to see how LLMs that are trained on their data perform vs. LLMs that are trained on hand-crafted datasets.

My overall impression is that the authors should focus on either the resource side of their work, or the empirical side. In its current state, the lack of focus in combination with a (at times) poor representation makes the paper appear inconsistent and at times hard to follow.

**Questions:**

My suggestion for improvement would be to include more details about the Query Bootstrapping method and the metrics reported in the empirical section. I am confident that the language issues could be easily resolved as well (maybe with the help of an LLM even).
My main issue, however, is the unclear focus of the paper. For it to be a good resource/method paper, the data collection process should be described in more detail. For it to be a good empirical study paper, the experiments should reflect the argument of automating the dataset collection process vs. manual creation of a dataset. Unfortunately, this would require major revisions and potentially additional work. (The result however, might be two good papers (one with focus on the data, and another one with focus on the empirical evaluation), as the underlying questions are relevant and interesting)

---

> ### Author Response · Authors · 2024-11-21
>
> We greatly appreciate your valuable and insightful comments, which have been highly instructive. We are revising the manuscript in the following areas:
>
> 1. We will carefully revise the manuscript to improve clarity and readability, ensuring that the language issues do not detract from the quality of the paper.
>
> 2. In response to your suggestions regarding the presentation of results, we will revise all relevant sections to improve clarity and ensure better comprehension. Specifically, we will provide a more detailed explanation of the emerging QuRating metric, including its calculation method and significance, to enhance the reader's understanding of the experimental outcomes. Furthermore, with respect to the reported performance improvements (represented by 'points'), we will explicitly clarify the meaning of these 'points' in both the tables and the discussion section, providing the necessary background and context.
>
> 3. We understand the reviewer’s concern regarding the lack of clarity in the focus of the manuscript. To address this, we plan to improve the manuscript’s structure and emphasis to better highlight its key contributions. Specifically, we will clarify the central argument and how the two aspects of the study, dataset collection automation and empirical evaluation, are interrelated. By doing so, we will more clearly distinguish between the two parts, ensuring a more cohesive narrative throughout the manuscript.
>
> We sincerely thank you once again for your thoughtful feedback, which has been instrumental in guiding our revisions and improving the overall quality of our manuscript.

---

### Official Review · Reviewer_KvvZ · 2024-11-04

**Soundness:** 3
**Presentation:** 2
**Contribution:** 2
**Rating:** 6
**Confidence:** 3

**Summary:**

This paper provides a method called Retrieve-from-CC to curate domain specific data to train large language models. It uses a two phased approach where initial query keywords are given by humans and a LLM is used to generate queries which are fed to retriever (BM25) to gather data that is relevant for a specific domain. Authors publish a benchmark, Retrieve-Pile, that is covering four domains that includes sciences, humanities, Social Sciences and Miscellaneous.  Paper shows that using this data set helps in improving the performance on some of the mathematical benchmarks along with standard language benchmarks.

**Strengths:**

Peper described a method for collecting domain specific training data without manual intervention, which is beneficial for makings LLMs perform better for domains of interest and making them more suitable for practical applications for a domain of interest. Method being automatic and showing improvements over the base LLMs is promising and can be beneficial in gathering large data that LLMs need for their training. Empirical results show that data generated results in significant improvements using it in LLM training. Quality metric also show that curated data is of good quality. Data pile created shows improvements during per-training as well as further training existing open models, which shows that generated data is of good quality.

**Weaknesses:**

I see some places Knowledge-Pile is used without it being talked about anywhere. I guess its a typo instead of Retrieve-Pile it is used in evaluation section, tables, figures etc. This needs to be corrected. Lot of places I see space missing after Retrieve-Pile and other typos needs to be corrected as well.

**Questions:**

Which LLM is used for query generation module? Did you see any difference in the queries generated with various LLMs ?
Assumption is that LLMs are not great at domain specific tasks, how does that impact your automatic query generation. Did you analyze the quality of queries generated for the domains?
Is it the same model that is used for query generation or a bigger LLM is used ?

---

> ### Author Response · Authors · 2024-11-21
>
> We sincerely thank the reviewer for their thoughtful comments on our paper. We will address your questions in detail below:
>
> 1. Regarding the weaknesses, we apologize for the typographical errors present in the paper, which have been corrected in the revised version.
>
> 2. We use the LLaMA2 13B model to generate all queries, including both questions and thoughts. We apologize for not providing a detailed explanation of the data collection process in the previous version. This content will be included in the revised version.
>
> 3. In this paper, our primary goal is to collect high-quality domain-specific data, which makes the choice of LLMs relatively independent. However, the use of different models for query generation in retrieval is an area for promising future work, which seems to be of significant importance.
>
> 4. In Retrieve-from-CC, the method does not rely on the model's performance on task-specific tasks in this field, but rather on its ability to generalize. For instance, the LLaMA 13B chat model, despite achieving only 54.8% accuracy on the MMLU benchmark, enables the smaller LLaMA 7B model to achieve or even surpass this performance when used to build Retrieve-Pile.
>
> 5. Due to the hallucination problem in large models, we found that the queries generated by these models are often inaccurate. However, since we use the queries generated by the large model solely to retrieve data from a public corpus, the quality of the queries only minimally affects the final retrieved corpus.
>
> 6. In the query generation process, we use the LLaMA2 13B model to generate the queries. During the training phase, to optimize training costs, we utilize the LLaMA2 7B model.
>
> We greatly appreciate your valuable feedback, which will undoubtedly help us improve the quality of our work.

---

### Official Review · Reviewer_8mWq · 2024-11-04

**Soundness:** 2
**Presentation:** 3
**Contribution:** 3
**Rating:** 6
**Confidence:** 3

**Summary:**

This paper investigates the problem of bootstrapping domain knowledge from general public corpora in order to reduce cost and time for manual data collection of domain-specific corpora. Using manually defined seed the presented approach, Retrieve-from-CC, first identifies seed examples in a large input corpus using BM25. For every retrieved record a LLM generates questions and responses are augmented by Chain of Thought sequences. After a quality filtering step the approach outputs a domain-specific dataset.

**Strengths:**

Generating high quality datasets for augmenting the capabilities of LLMs into less covered domains is high relevant for the open-source as well as professional community. Data is the key for LLM success and this paper aims to present contribute for this matter.

**Weaknesses:**

*Approach*:
- seeds: A downside of the approach is that it needs good seeds as input. At the same time, general-domain knowledge bases (dbpedia, yago) cover almost all domains to some degree. While the data goes beyond simple key phrases, the domains are probably only covered partially. The authors should consider leveraging this partial knowledge for generating input data for the bootstrapping phase. This drops the manual input requirement and might improve the dataset further.

*Evaluation*:
- My biggest criticism of the paper is that the author didn't compare against domain-specific LLMs. The question: "How does a LLM trained over a corpus generated with Retrieve-from-CC compare against a domain-specific LLMs?" is highly relevant for this paper and is not answered. For instance, one could compare against LLMs from the [Open Medical-LLM Leaderboard](https://huggingface.co/spaces/openlifescienceai/open_medical_llm_leaderboard). There are probably other such domain-specific resources.


* Misc: Typo generted in Figure 2

**Questions:**

Beyond the tasks you evaluated were there any performance changes after you further trained the LLM with Retrieve-Pile?

---

> ### Author Response · Authors · 2024-11-21
>
> Tanks for your helpful review, we will discuss these question below.
>
> 1. Regarding the scope of seeds, we acknowledge that seed data plays a significant role in our approach. Exploring a broader knowledge base as a retrieval seed can yield more diverse and higher-quality data, representing an important direction for future research. In this paper, however, we aim to demonstrate that leveraging the generalization ability of a large language model enables the collection of data relevant to a specific field using only a few keywords. Corresponding experiments have shown that the data collected using our method improves performance on domain-specific benchmarks. Additionally, enhancing the scope and quality of seeds warrants further attention. We plan to enhance this approach in future work to generate more generalized and higher-quality data.
>
> 2. We apologize for omitting this comparative analysis in the earlier version. In the revised manuscript, we will include additional experiments to compare the performance of our model with domain-specific models.
>
> |                         | Clinical KG | Medical Genetics | Anatomy | Pro Medicine | College Biology | College Medicine |
> |-------------------------|-------------|------------------|---------|--------------|-----------------|------------------|
> | Mistral 7B Instruct     | 62.9        | 57               | 55.6    | 59.4         | 62.5            | 57.2             |
> | BioMistral 7B（3-shot） | 60.9        | 61.7             | 49.6    | 55.1         | 56.9            | 55.5             |
> | BioMistral 7B SLERP     | 63.1        | 63.3             | 49.9    | 57.4         | 63.4            | 57.8             |
> | Mistral-QoC             | 70.57       | 74               | 60.74   | 58.82        | 77.08           | 67.63            |
>
> As shown in the table above, compared to BioMistral, Mistral-QoC outperforms BioMistral across all MMLU-related subsets. This may be attributed to two factors: firstly, the retrieval of CC allows for the collection of a large amount of data at a minimal cost; secondly, most medical domain data is not concentrated, and BioMistral only utilizes a limited subset of PMC Open Access data, which restricts its sources and volume.
>
> 3. In our evaluated benchmark, BBH is a highly complex and general reasoning benchmark that is less directly aligned with our approach and seed information. Nevertheless, it demonstrated a significant improvement.
>
> Thank you for your review. If you have any further questions, we will respond promptly.

---

> > ### Comment · Reviewer_8mWq · 2024-11-29
> >
> > I appreciate the author's additional contributions. The revisions to the Evaluation section effectively addressed enhanced the overall quality of the manuscript.

---

### Meta-Review · Area_Chair_FXVZ · 2024-12-19

**Metareview:**

**Summary**

Finding reliable data for LLMs is the challenge this paper focus on. The authors propose Retrieve-from-CC, a data collection pipeline consisting of a a) query generation process based on existing language models and b) a document retrieval process based on the generated queries. LLMs are used as query expansion.

**Strengths**

The goal of the paper is extremely important as find reliable data is one of the big challenges in improving LLMs

**Weaknesses**

- An evaluation of the method in term of quality of the data does not seem to be provided.
- The retrival cretiria should be expanded and explained.

**Final remarks**

The paper fails to convince reviewers that have interacted with the authors.

**Additional Comments On Reviewer Discussion:**

The discussion has been fruitful and the position of the authors and of the reviewers have been clarified.

---

### Decision · Program_Chairs · 2025-01-22

Reject